# Sacrospinous Hysteropexy Versus Prolapse Hysterectomy with Apical Fixation: A Retrospective Comparison over an 18 Year Period

**DOI:** 10.3390/jcm12062176

**Published:** 2023-03-10

**Authors:** Greta Lisa Carlin, Sören Lange, Christina Ziegler, Florian Heinzl, Barbara Bodner-Adler

**Affiliations:** Department of General Gynecology and Gynecolgic Oncology, Medical University of Vienna, 1090 Vienna, Austria

**Keywords:** sacrospinous hysteropexy, vaginal hysterectomy, pelvic organ prolapse, prolapse surgery

## Abstract

Background. Pelvic organ prolapse (POP) is a common health problem, with a high lifetime risk for prolapse surgery. Uterine-preserving procedures such as vaginal sacrospinous hysteropexy (SSH) have become an increasingly utilized surgical option for the primary treatment of POP. We wanted to evaluate peri- and postoperative outcome parameters of SSH as an alternative to vaginal hysterectomy with apical fixation. Methods. A retrospective cohort study was conducted (2003–2021). All patients who underwent primary SSH (study group) for symptomatic POP were matched 1:1 by age and BMI with patients who underwent primary prolapse hysterectomy with apical fixation (control group). Results. A total of 192 patients were included with 96 patients in the each of the SSH and hysterectomy groups. There were no statistically significant differences in baseline characteristics. The SSH group show a significantly shorter mean surgery time (*p* < 0.001), significantly fewer hospitalization days (*p* < 0.001), and significantly less intraoperative blood loss (*p* = 0.033) in comparison to the control group. Neither group had any intraoperative complication, or an intraoperative conversion to other surgical management options. No statistically significant difference was found in postoperative complications as categorized by the Clavien–Dindo classification or in postoperative urogynecological issues (UTI, de-novo, incontinence, residual urine, voiding disorders). Through log regression, none of the confounding factors such as age, BMI, or preoperative POP-Q stage could be identified as independent risk factors for the occurrence of postoperative complications. Conclusions. Our results confirm that a uterus-preserving technique has many benefits and, thus, should be considered as an additional intermediate step in a long-term treatment plan of pelvic organ prolapse.

## 1. Introduction

Pelvic organ prolapse (POP) is a common health problem and due to the world’s increasingly older population, its incidence is rising [1]. Of symptomatic POP cases, around 3–6% are associated with symptoms connected to a decisive lower quality of life, including bladder and bowel issues as well as sexual dysfunction [2,3]. Therefore, these women should be offered feasible, safe, and functional treatment options. Individual treatment plans based on patient’s expectations, wishes, and necessities should be chosen. Treatment options can range from conservative pessary application to various surgical repair methods [4,5], with the lifetime risk for POP surgery being around 12–19% in the general female population [6,7,8]. There are various options for the surgical treatment of uterine prolapse, with vaginal hysterectomy still being the standard procedure in many countries.

However, various studies have also shown that prolapse hysterectomy independently increases the risk for a subsequent prolapse or recurrence, even in nulliparous women [9,10,11,12]. To prevent subsequent POP occurrence, there are many different procedures that can be performed to suspend the vaginal stump during hysterectomy that have been postulated. Uterosacral ligament suspension (USLS) and sacrospinous ligament fixation are often recommended at the time of vaginal hysterectomy, as suturing the vaginal cuff to either cardinal or uterosacral ligaments is hypothesized to counteract the decrease in vaginal support by the disruption of the ligament suspension [13,14].

However, as women have expressed more and more the desire to keep their uterus, procedures that preserve the uterus such as vaginal sacrospinous hysteropexy (SSH) are becoming more popular as primary surgical treatment option for POP [4]. There are not many studies that compare vaginal hysterectomy (vag. HE) with uterus-preserving procedures and data on long-term outcomes are spare. Sacrospinous hysteropexy is an uterine-preserving, well-established surgical procedure that gained popularity after the publication of the SAFE U (sacrospinous fixation versus vaginal hysterectomy in treatment of uterine prolapse ≥2) trail conducted by the Dutch research group of Detollenaere and Schulten et al. about its successful use in uterine preservation in prolapse surgery [15,16]. Other studies also researched surgical outcomes after prolapse surgery and reached the conclusion that compared with vag. HE, SSH has similar, or possibly even better, outcomes [17,18,19].

Sacrospinous hysteropexy is increasingly more performed for uterovaginal prolapse repair and, currently, it is the most commonly performed uterine-preserving surgical technique at our institution in Austria. The aim of this study was to evaluate peri- and postoperative outcome parameters of uterine-preserving sacrospinous hysteropexy as an alternative treatment option to vaginal prolapse hysterectomy with apical fixation.

## 2. Materials and Methods

This was a retrospective cohort study. All patients who underwent either primary SSH (study group) or primary prolapse hysterectomy with apical fixation by uterosacral ligament suspension (USLS) (control group) for symptomatic POP at our tertiary hospital from 2003 to 2021 were eligible for inclusion and their electronic hospital chart was consulted. It is noteworthy that the percentage of SSH increased significantly over the last 5 years at our institution, even surpassing prolapse hysterectomy with apical fixation, which was the dominant procedure for primary POP surgery at our institution the years before. Nevertheless, of the 1610 patients who underwent a surgical prolapse repair (prolapse hysterectomy or uterus-preserving prolapse surgery) due to symptomatic POP in the last 18 years at our institution, only a small percentage of these patients (*n* = 110) underwent a vaginal SSH as an index operation. The reason for that is—as mentioned above—the trend towards uterus-preserving prolapse surgery in the last 5 years, but not earlier.

Exclusion criteria were as follows: emergency cases, treatment for non-urogynecological issues, malignant results, history of previous pelvic floor surgeries. Additionally, 18 patients were excluded due to an incomplete data set. This resulted in a study population of 96 patients. These patients were matched 1:1 by age and BMI with patients who underwent vag. HE with USLS during the same time period. Institutional review board approval was obtained for this study.

Demographic and clinical information including confirmation of pelvic organ prolapse (all included patients suffered from a symptomatic POP with a POP-Q stage >2, which was evaluated with the ICS—POP Q system [20]), as well as assessment of pelvic floor muscles in accordance to the Oxford scale [21] and operative records were obtained from the included patients’ electronic hospital chart. All patient records were anonymized and de-identified prior to analysis. Postoperative complications recorded in the patients charts during their postoperative stay and during postoperative follow-up clinical controls were assessed through the standardized classification of surgical complications according to Clavien–Dindo [22].

### 2.1. Description of the Surgical Procedure

All patients were operated by the vaginal route. All preparations occurred under aseptic conditions and standardized single-shot antibiotic prophylaxis was administered before incision.

All procedures were performed by a surgical core team consisting of 3 consultants for the urogynecology and reconstructive pelvic floor surgery division and a uniform technique was used as described.

#### 2.1.1. Sacrospinous Hysteropexy

The surgical procedure of SSH performed at our institution is conducted as follows: high posterior colpotomy is made towards the posterior cervix, then blunt preparation towards the right ischial spine to visualize the right sacrospinous ligament. The posterior side of the cervix is joined to the right sacrospinous ligament with two late-absorbable sutures (PDS sutures–0) at least 2 cm medial to the ischial spine. This suture is then passed through the posterior cervical wall, but not yet knotted. First, the colpotomy is closed via absorbable sutures (2/0 vicryl, Ethicon, Sommerville, NJ, USA). Where additional procedures such as anterior and/or posterior colporrhaphy were needed, they were performed at this stage. Only after are the pre-laid fixation sutures tied, whereby the cervix comes to lie about 4–6 cm cranial of the level off the vulva towards the sacrospinous ligament, without the cervix abutting the sacrospinous ligament, but so that the knot on the side of the ligament slips off slightly, in order to avoid the occurrence of necrosis or the cutting of the thread into the ligament. It, thus, follows closely the surgical procedure of SSH previously described by Schulten et al., however, in contrast, no surgical devices were used [16].

#### 2.1.2. Vaginal Hysterectomy

The vaginal wall around the cervix is circumcised. After bladder and bowel dissection, the peritoneum (anterior and posterior) is opened. The uterosacral ligaments on both sides are identified, ligated, and transected. The uterus is then removed step-by-step using several clamps and sutures. After removal of the uterus, the pedicles are inspected for hemostasis. Additionally, a uterosacral ligament suspension (USLS= attachment of the uterosacral ligaments to the vaginal vault with two delayed-absorbable sutures in PDS 2.0, Ethicon) is performed to suspend the vault.

The sutures are placed as high as possible on the visible part of the clamped and ligated uterosacral ligament bilaterally: first through the vaginal epithelium at full thickness, then through the clamped uterosacral ligament left side, as high as possible, then several stitches through the peritoneum, and then again through the right uterosacral ligament and posterior wall of the vaginal epithelium—but not yet knotted. A second row, situated more proximal to the first, is also laid out but not yet knotted. Where additional procedures such as anterior colporrhaphy are needed, they are performed at this stage. Only after are the pre-laid high USLS fixation sutures tied. Thereafter, should a posterior colporrhaphy be needed, it is performed.

Afterwards, a cystoscopy with 33% glucose is performed in all cases to exclude any injuries of the ureter, detect ureter kinking, and detect an adequate urine stream from both ostia. Where additional procedures such as anterior and/or posterior colporrhaphy are needed, they are performed at this stage.

### 2.2. Primary Outcome Parameters

Our primary outcome parameters were defined as follows:Operating time, defined as starting and ending with the patients’ anesthesia in the operating theatre;Intraoperative blood loss, as documented by the surgeons immediately post intervention;Intraoperative complications, defined as any deviance from standardized procedure documented by the surgeon;Intraoperative conversion to other surgical management options;Postoperative complications as classified by the Clavien–Dindo classification [22]

As well as postoperative, de-novo incontinence, or voiding disorders; pathological residual urine; vaginal infection; infection of the wound; or profuse bleeding that required blood transfusions; and pain.

### 2.3. Statistical Analysis

Statistical analysis was performed with R (R Core Team 2022) and packages Barnad [23], flextable [24], and ggplot2 [25]. Numerical data are described via medians and interquartile ranges. The potential differences in such variables were examined with the Wilcoxon rank sum test. Categorical data are described via absolute frequencies. For group comparisons concerning variables with two categories, we used Barnard’s test. In case of three or more, we chose Pearson’s χ^2^-test. *p*-values < 0.05 were considered statistically significant (two-sided).

## 3. Results

A total of 1610 patients were assessed for eligibility. Of these, 1418 patients were excluded either for not meeting the inclusion criteria or because of an incomplete data set. Therefore, 192 patients were included: 96 patients in each of the study and control groups (Figure 1).

### 3.1. Patients’ Characteristics

Baseline characteristics (age, BMI, menopausal stage, parity, delivery mode, etc.) between the study and control group did not show any statistically significant differences.

Furthermore, there was no statistically significant difference in preoperative POP-Q stage between the groups. Additionally, no statistically significant difference regarding preoperative health issues such as cardiovascular disease, renal diseases, diabetes, or poly-medication could be found (Table 1).

### 3.2. Primary Outcomes

The SSH group shows a significantly shorter mean surgery time with 65 vs. 95 min (*p* < 0.001) (Figure 2); significantly fewer hospitalization days with 2.94 vs. 4.87 days (*p* < 0.001) (Figure 3), and significantly less intraoperative blood loss with 120.00 mL vs. 186.18 mL (*p* = 0.033) (Figure 4) in comparison to the control group (Table 2).

### 3.3. Intraoperative and Postoperative Complications

No intraoperative complication were reported in either group, nor was an intraoperative conversion to another surgical method documented. As for postoperative complications, three patients experienced a grade 3b complication as categorized by the Clavien–Dindo classification (Figure 5), however, this did not constitute a statistically significant difference between the groups (Table 3). These three 3b complications consisted of one patient in the study group with postoperative bleeding that was treated in the operating theatre, and two patients in the control group, one with a vesicovaginal fistula and one with an apex infection, that had to be treated in the operating theatre.

Regarding postoperative issues such as de-novo incontinence or voiding disorders, residual urine, UTIs, vaginal infection or infection of the wound, bleeding, required blood transfusions, and pain, the control and study group show no statistically significant differences (Table 3).

While not statistically significant, it is interesting to note that there were 12 patients with a postoperative residual urine issue in the control group, compared to 5 patients in the study group.

### 3.4. Logistic Regression

Logistic regression of postoperative complications was conducted in the whole patients’ collective depending on the group affiliation with vaginal hysterectomy as baseline (Table 4) and in the study group (Table 5). None of the confounding factors such as age, BMI, or preoperative POP-Q stage could be identified as independent risk factors for the occurrence of postoperative complications.

## 4. Discussion

There is only a limited number of studies that compare vaginal hysterectomy to uterus-preserving procedures and, thus, data on long-term follow-ups after SSH are still not available. Only a handful of studies have reported their one year and/or five year follow-up results [15,16,26], and even they focus on the peri- and short-term postoperative results.

One randomised controlled trial, published in 2010 by Dietz et al., analyzed 66 patients, where 33 cases underwent SSH and 31 women received a prolapse hysterectomy with or without apical fixation [26]. Their results show a significant reduced length of time to return to work in the SSH group (43 versus 66 days, *p*  =  0.02) as well as significantly shorter hospitalisation in the SSH group (3 days in the SSH group vs. 4 days in their vaginal hysterectomy group; *p* = 0.03). Furthermore, no bladder or rectal injuries, as well as no blood transfusions or intensive care admissions, were reported in either group. These postoperative outcomes, as well hospitalisation rates, align with our findings.

Furthermore, the SAFE U trial [15] reported that their operative time consisted of 59 min in the study (SSH) vs. 72 min in the control group (vag. HE). The statistically significant shorter operating time is also in line with our findings. However, in contrast to our findings, no difference regarding intraoperative blood loss could be detected between their two groups. Nevertheless, our findings show a statistically significant diminished intraoperative blood loss in the SSH group (120.00 mL vs. 186.18 mL; *p* = 0.033).

Additionally, the reported complication in their postoperative period showed no significant difference between the study and control group, which corresponds to our findings. However, the authors mentioned five serious adverse events occurring during hospital stay: two after vaginal hysterectomy (one death after paralytic ileus and aspiration pneumonia and multiple organ failure, and one atrial fibrillation requiring cardioversion) and three after sacrospinous hysteropexy (one stroke, one pneumonia, and one anaphylactic reaction to prophylactic antibiotics).

However, it must be taken into consideration that our study was conducted as a retrospective analysis and that, thus, different primary outcome parameters were used. Therefore, a complete comparison between outcome variables is not possible. Furthermore, it should be mentioned that surgical techniques differ between reported studies, making the comparison of results even harder. To state only one example, our institution uses long-term absorbable PDS sutures for the fixation suture during SSH, while Dietz et al., as well as Detollenaere et al., performed the same suture with non-resorbable Prolene suturing material [15,26]. It is unclear if and how this different suturing material, as well as other differences in surgical technique between studies, might have impacted the results. It is also unclear how this might affect the long-term outcome.

Nevertheless, suitable treatment options for POP are an important gynecological issue, especially as the global demand for health care services related to pelvic floor issues is expected increase in accordance to ageing demographic and easier health care access [27]. As such, demand for options of surgical treatments of POP is likely to increase as well. Therefore, it is important to consider each woman’s overall wishes when making treatment decisions and offer proper counselling, as well as tailor treatment plans in accordance to patients’ individual expectations and needs. As sacrospinous hysteropexy gives us a useful additional intermediate step in a long-term treatment plan of pelvic organ prolapse, it should be readily included as a further operative technique.

### Strengths and Limitations

We are aware of the limitations of our study. Since this was a retrospective cohort study, there are limits in determining a cause-and-effect relationship. Only associations between operation method (SSH vs. vag. HE) and peri- and postoperative complications, operating time, and hospitalization can be made. Furthermore, also due to the design of this study, sadly no information on postoperative anatomical outcome is possible, even though we are aware of the high clinical significance of such data. In addition, while all counseling at our outpatient clinic is performed in a non-directive manner, due to the study design, we cannot completely rule out if different counseling occurred and, thus, selection bias, nor can a consistency in pre-operative assessment be guaranteed. In regards to long-term follow-up, we plan to conduct a follow-up study with telephone interviews with the patients’ to inquire after their subjective outcome and satisfaction.

On the other hand, strengths in our study are that this study includes a large population of 96 SSH patients that were well-matched by age and BMI with patients receiving a primary vaginal HE with apical suspension. Additionally, there were no significant differences in the baseline characteristics of the groups and all our preoperative clinical examinations, as well as surgical procedures, were all performed by a small group of experienced urogynecological specialists. This led to a consistency of preoperative measurements and a standardized route of surgery and material during the operations. Thus, our study significantly enlarges the still scarce literature comparing these two prolapse operation methods.

## 5. Conclusions

Our results confirm that the uterus-preserving surgical alternative shows several benefits such as a shorter operating time and a reduced hospital stay, as well as diminished intraoperative blood loss. As sacrospinous hysteropexy gives us an additional intermediate step in a long-term treatment plan of pelvic organ prolapse, it might be considered and offered as a surgical treatment option to our patients. However, long-term follow up data are still very much needed. We are currently working on conducting telephone interviews with the patients to assess their subjective outcome and hope to be able to publish these results in the future.

## Figures and Tables

**Figure 1 jcm-12-02176-f001:**
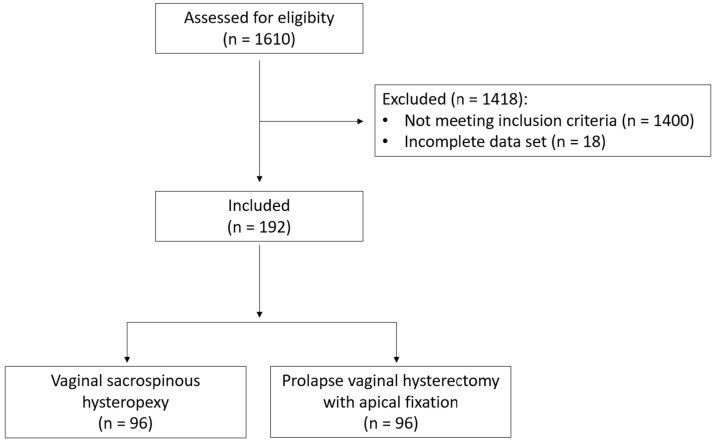
Flow of patients through the study.

**Figure 2 jcm-12-02176-f002:**
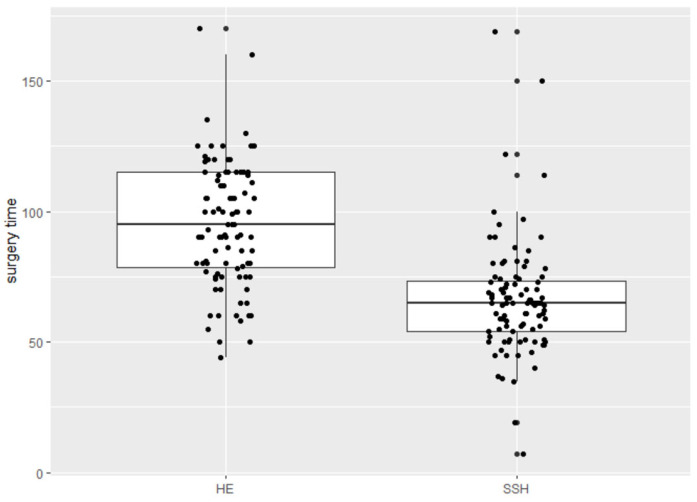
Surgery time in minutes.

**Figure 3 jcm-12-02176-f003:**
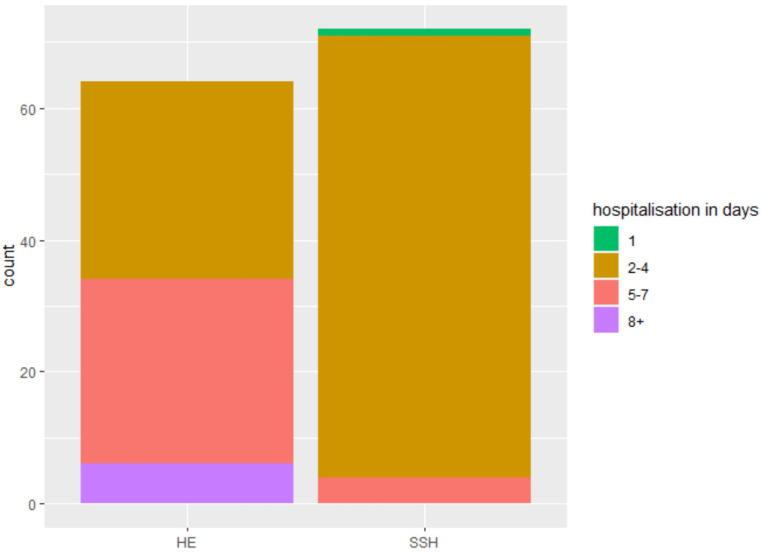
Hospitalization days.

**Figure 4 jcm-12-02176-f004:**
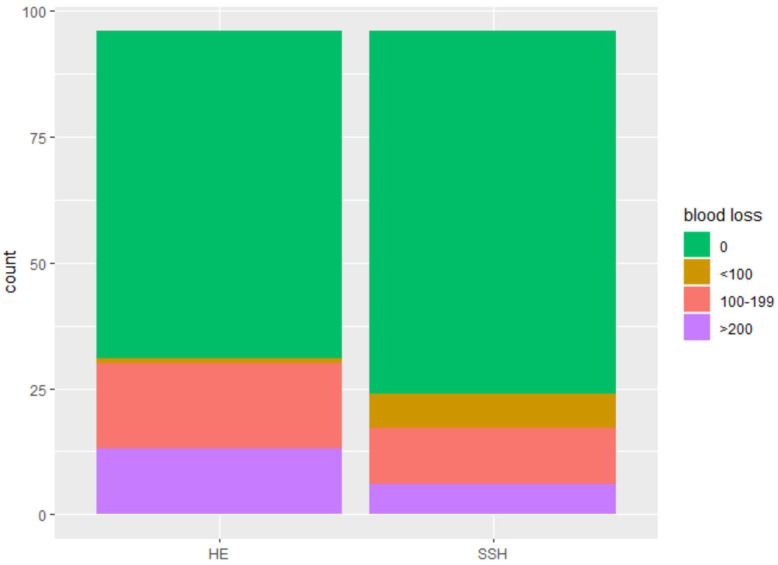
Intraoperative blood loss in milliliters.

**Figure 5 jcm-12-02176-f005:**
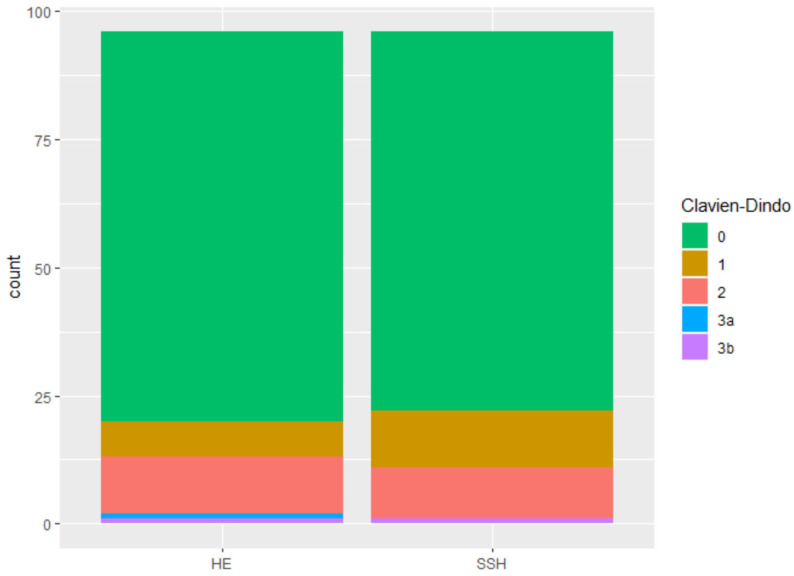
Postoperative complication after Clavien–Dindo classification.

**Table 1 jcm-12-02176-t001:** Patients’ baseline characteristics.

Variable	All	HE	SSH	*p* Value	Remark	Test
**count**	192	96	96			
**age**	58.91 (50.83–66.76)	58.81 (50.83–66.80)	58.91 (50.83–66.76)	0.8334	NAs: 0 in group HE, 0 in group SSH	Wilcoxon rank sum test with continuity correction
**BMI**	26.22 (23.33–29.86)	26.95 (24.4–29.71)	25.65 (22.97–30.03)	0.1335	NAs: 8 in group HE, 29 in group SSH	Wilcoxon rank sum test with continuity correction
**preoperative POP-Q stage**	3 (2–3)	3 (2–3)	3 (2–3)	0.6102	NAs: 0 in group HE, 0 in group SSH	Wilcoxon rank sum test with continuity correction
**preoperative Oxford scale**	0 (0–2)	0 (0–0)	0 (0–3)	0.0002	NAs: 0 in group HE, 0 in group SSH	Wilcoxon rank sum test with continuity correction
**preoperative cardiovascular issues**				0.2554		Barnard’s Unconditiol Test
no	145	76	69			
yes	47	20	27			
**preoperative diabetes**				0.5289		Barnard’s Unconditiol Test
no	179	91	88			
yes	13	5	8			
**preoperative renal insufficiency**				1		Barnard’s Unconditiol Test
no	191	96	95			
yes	1	0	1			
**preoperative poly-medication**				0.5305		Barnard’s Unconditiol Test
no	140	74	66			
yes	31	14	17			
**preoperative menopausal stage**				0.4414		Pearson’s Chi-squared test with Yates’ continuity correction
premenopausal	28	13	15			
perimenopausal	20	7	13			
postmenopausal	105	53	52			
**preoperative nicotine consumtion**				0.9846		Barnard’s Unconditiol Test
no	100	44	56			
yes	18	8	10			
**preoperative parity**				0.624		Pearson’s Chi-squared test with Yates’ continuity correction
0	12	4	8			
1	35	17	18			
2	85	41	44			
3	38	21	17			
4+	22	13	9			

**Table 2 jcm-12-02176-t002:** Primary outcomes.

Variable	All	HE	SSH	*p* Value	Remark	Test
**surgery time (min)**	75 (61–100)	95 (78.75–115)	65 (54–73.25)	0.0000	NAs: 0 in group HE, 0 in group SSH	Wilcoxon rank sum test with continuity correction
**postoperative hospitalisation (days)**				0.0000		Pearson’s Chi-squared test with Yates’ continuity correction
1	1	0	1			
2–4	97	30	67			
5–7	32	28	4			
8+	6	6	0			
**blood loss (mL)**				0.0332		Pearson’s Chi-squared test with Yates’ continuity correction
0	137	65	72			
<100	8	1	7			
100–199	28	17	11			
>200	19	13	6			

**Table 3 jcm-12-02176-t003:** Postoperative complications and issues.

Variable	All	HE	SSH	*p* Value	Remark	Test
**count**	192	96	96			
**postoperative complications**				0.1465		Barnard’s Unconditiol Test
no	156	74	82			
yes	36	22	14			
**postoperative UTIs**				1.0000		Barnard’s Unconditiol Test
no	182	91	91			
yes	10	5	5			
**postoperative bloodtransfusion**				0.2105		Barnard’s Unconditiol Test
no	190	94	96			
yes	2	2	0			
**postoperative bleeding**				0.5319		Barnard’s Unconditiol Test
no	186	94	92			
yes	6	2	4			
**postoperative pain**				1.0000		Barnard’s Unconditiol Test
no	186	93	93			
yes	6	3	3			
**postoperative residual urine**				0.0831		Barnard’s Unconditiol Test
no	175	84	91			
yes	17	12	5			
**postoperative bowel issue**				0.5319		Barnard’s Unconditiol Test
no	186	92	94			
yes	6	4	2			
**postoperative wound infection**				1.0000		Barnard’s Unconditiol Test
no	190	95	96			
yes	2	1	1			
**postoperative fistula**				0.5289		Barnard’s Unconditiol Test
no	191	95	96			
yes	1	1	0			
**Clavien-Dindo score**				0.7425		Pearson’s Chi-squared test with Yates’ continuity correction
0	150	76	74			
1	18	7	11			
2	21	11	10			
3a	1	1	0			
3b	2	1	1			

**Table 4 jcm-12-02176-t004:** Logistic regression. Postoperative complications in the whole collective depending on the group affiliation (vag. HE as baseline): age, BMI, preoperative POP-Q stage. *p* = 0.4978; residual deviances: 146.44 for the null model vs. for the full 143.06 model.

	Estimate	Standard Error	z Value	Pr (>|z|)
**(Intercept)**	−0.845	1.852	−0.456	0.6482
**SSH group**	−0.655	0.430	−1.521	0.1282
**age**	0.013	0-017	0.730	0.465
**BMI**	−0.010	0.050	−0.195	0.8451
**preoperative POP-Q stage**	−0.314	0.483	−1.651	0.5152

**Table 5 jcm-12-02176-t005:** Logistic regression. Postoperative complications in the study group (SSH): age, BMI, preoperative POP-Q stage. *p* = 0.7821; residual deviances: 64.648 for the null model vs. for the full 63.343 model.

	Estimate	Standard Error	z Value	Pr (>|z|)
**(Intercept)**	0.235	2.739	0.086	0.9317
**age**	−0.019	0.026	−0.746	0.4556
**BMI**	0.013	0.072	0.184	0.8537
**preoperative POP-Q stage**	−0.506	0.702	−0.720	0.4717

## Data Availability

The data presented in this study are available on request from the corresponding author. The data are not publicly available due to privacy restrictions.

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
