# Peer review of "Sacrospinous Hysteropexy Versus Prolapse Hysterectomy with Apical Fixation: A Retrospective Comparison over an 18 Year Period"

_jcm, 2023, doi:10.3390/jcm12062176_

Round 1
Reviewer 1 Report
The authors presets a comparison between the outcomes of sacrospinous hysteropexy versus hysterectomy with apical fixation in the management of POP.
- Non-randomised trial which could lead to bias in selecting the treatment option offered for the patient.
- Non-randomised trial which could lead to bias in selecting the treatment option offered for the patient.
- There was no consistency in pre-operative assessment.
- Authors should specify inclusion and exclusion criteria in detail
The ICS POP-Q system uses a moving structure, the hymen, as the reference point to quantify pelvic organ descent, which may not be optimal for this study. Ultrasound is used increasingly in the evaluation of women with pelvic floor dysfunction. One of the main applications of translabial ultrasound imaging is in the quantification of POP, which is useful in both clinical audit and research. One of the fundamental questions in the assessment of pelvic organ prolapse is ‘what is significant pelvic organ descent? Based on current evidence, it is apparent that assessment of the pelvic floor, clinically and using imaging should be an integral part of POP assessment. It is important not only in preoperative counseling but also in surgical planning, for example in the selection of patients. Usually, prolapse is staged according to POP-Q coordinates (cfr.: , , . Correlation of symptoms with degree of pelvic organ support in a general population of women: What is pelvic organ prolapse. Am J Obstet Gynecol 2003; 189: 372– 379.) resulting in large proportions of the female population being diagnosed erroneously.
Authors should specify secondary outcomes such as recurrence and an atomical failure
Reviewer 2 Report
This study adds to the existing literature supporting the peri-operative advantages of transvaginal Hysteropexy (SSH) compared to transvaginal Hysterectomy with high USL suspension (Hx-USLS). Known advantages of SSH include less operative time (here approx 1hr vs 1.5 hrs), less blood less (here 120cc vs 180cc -- which I would argue really is NOT a clinical significance with respect to clinical care/outcomes and is an "at best guess" by the surgeon anyway, not objectively measured), and fewer days in the hospital (ave 3 vs 5 -- which should be explained more since both of these procedures are commonly done on an outpatient basis; especially at many hospitals in the US). Although I appreciate that the authors were forthright and stating their aims of this 18 year study was to look at these peri-operative parameters, this is a missed opportunity to answer the more important question of long-term surgical outcome. The authors acknowledge that only few studies exist in our literature that attempt to look at comparative surgical outcomes and recurrence rates, noting mostly 1-year data only; one 5-year study was cited too. This is the TRUE variable needed in order to make informed recommendations to patients concerning methods of POP surgery.
Again, this DOES add to the limited data already in the literature about peri-op issues following these two procedures, but a follow-up publication is HIGHLY encouraged, looking about long-term POP recurrence between these two procedures. Just looking at peri-op issues, it's not really fair to make substantive recommendations about which procedure might be better. The conclusion needs to be absolutely specific and can not mislead the reader into thinking that SSH is overall "better".
There are a few English language mistakes/typos that need correcting
Questions within the methods that need clarification:
1. Did either surgical technique change over the 18 years?
2. SSH -- Done Unilaterally with 2 Delayed absorbable sutures. Tied down as final maneuver -- to approximate the apex 4-6cm cranial to the introitus. Was this tied down such that the cervix was ABUTTING the SSL? Doesn't sound as such with only 4-6cm cranial. With delayed absorbable suture, one would question the long-term success of this. Other studies have noted when permanent material is used in this way (prolene and/or mesh), leaving a "suture/mesh bridge" between the cervix and the SSL, that the outcome is good.
2. High USLS --- 2 sutures were placed --- was this done bilaterally? Two sutures bilaterally? --- were the sutures placed through the vaginal epithelium into the vagina? When needed, were additional ant/post repairs done PRIOR TO tying down the apical sutures. Again, was the goal 4-6cm cranial from the introitus, which would imply a "suture bridge" between the apex and the USL (which is not classically how it it designed).
3. WHY were 1418 patient excluded from this retrospective review. You just state that they didn't meet inclusion criterion, but you don't state what those criterion are. The vast majority of patients were excluded. How do you account for that? Out of 1610 pt who underwent these procedures, 1418 were excluded ---- leaving only 192 for your review. Although 192 is still a handsome number, it still needs to be explained WHY the vast majority of patients undergoing these procedures were excluded. And .... in that case, is it REALLY an 18-year study? What is the time span between the 192?
4. You state there were NO intraop complications or "conversions to other surgical managment options". Really? Were intraop complications an exclusion criterion?
The comments concerning the methods (surgical technique details) and English language I view as minor revisions
Again, I think this is an important topic, comparing SSH to Hx-USLS and it needs to be investigated further, but the real issue is one of long-term surgical success versus recurrence!
Reviewer 3 Report
I read with great interest this manuscript describing the comparaison between sacroscopinous hysteropexy and prolapse hysterectomy with apical fixation. This retrospective study covers a period of 18 years which is not common.
Overall the manuscript is well written, the introduction is clear, the methods and results are clearly described.
However, I've two major concern about the methodology. The population studied at the beginning is 1610 but 1400 patients did not meet inclusion criteria. Please provide more explanation about the fact that only 12.5% of the patients where included in this retrospective study. If this is because they where operated for prolapse by different approach (such as abdominal sacrocolpopexy), they should not be taking into account on the total population number.
One of the main limitation is the absence of follow-up and therefore the lack of comparison of POP recurrence between these two techniques. Did you try to follow the patients, in retrospective files, or by phone call or new clinical examination ? The median follow-up should be an important outcome because the period covered by the study is very long.
The authors states in the conclusion that uterine preservation should be considered in the long-term POP treatment plan but this cannot be support by their findings.
Round 2
Reviewer 1 Report
A sufficient revision
Reviewer 3 Report
No more comments.